# Safe Policy Optimization with Local Generalized Linear Function Approximations

**Akifumi Wachi**
IBM Research
akifumi.wachi@ibm.com

**Yunyue Wei**
Tsinghua University
weiyy20@mails.tsinghua.edu.cn

**Yanan Sui**
Tsinghua University
ysui@tsinghua.edu.cn

## Abstract

*Safe exploration* is a key to applying reinforcement learning (RL) in safety-critical systems. Existing safe exploration methods guaranteed safety under the assumption of regularity, and it has been difficult to apply them to large-scale real problems. We propose a novel algorithm, SPO-LF, that optimizes an agent's policy while learning the relation between a locally available feature obtained by sensors and environmental reward/safety using generalized linear function approximations. We provide theoretical guarantees on its safety and optimality. We experimentally show that our algorithm is 1) more efficient in terms of sample complexity and computational cost and 2) more applicable to large-scale problems than previous safe RL methods with theoretical guarantees, and 3) comparably sample-efficient and safer compared with existing advanced deep RL methods with safety constraints.

## 1  Introduction

Applying reinforcement learning (RL) to applications with unknown safety constraints is challenging as RL is inherently an exploratory process and requires agents to learn the whole environment first. Though there has been a surge of attempts to incorporate safety in RL [3, 6, 11, 16], it is essentially difficult for most of the algorithms to guarantee safety, especially in the exploration phase. If a single mistake could lead to catastrophic failures, conventional algorithms cannot be directly applied.

To guarantee safety while learning an environment, *safe exploration* problems have been actively studied as a sub-field of safe RL [17]. A mainstream safe-exploration method is to utilize a Gaussian process (GP, [24]) under problem settings in which the agent can observe only the safety function value of its current state. Since the agent cannot get any information on the neighboring states, it is necessary to assume desirable properties for the function. Specifically, Sui et al. [31] assumed that a safety function is Lipschitz continuous and has regularity that can be captured by appropriate kernel functions. However, these assumptions do not hold in many environments. For example, in the context of autonomous driving, the degree of safety varies steeply depending on the presence or absence of other vehicles and pedestrians. Also, previous work on GP-based safe exploration suffers from inconsistency between their theoretical guarantees and computational cost. Previous work has proved the completeness of the predicted safe region [34] and the optimality of the acquired policy [36], *after a large number of samples*. However, the computational cost of GP is known to be large. Hence, it is hard to achieve theoretically guaranteed performance in a large environment.

Based on the fact that robots are equipped with sensors, it is more reasonable to formulate the problem assuming that robots can obtain "feature vectors" for inferring the degree of safety of the

35th Conference on Neural Information Processing Systems (NeurIPS 2021).

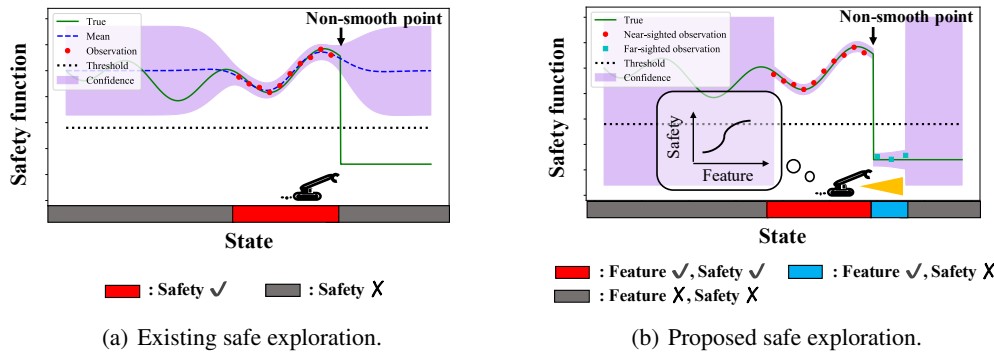

(a) Existing safe exploration.  (b) Proposed safe exploration.

Figure 1: (a) Previous GP-based safe exploration under regularity assumptions, which fails to handle non-smooth changes. (b) Our proposed feature-based safe exploration. In this problem setting, an agent predicts safety function value while using feature vector obtained by far-sighted observations.

(observable) states of their neighbors. This problem formulation is realistic for real-world robots, and it also allows us to deal with safety that changes steeply, which previous research could not handle (see Figure 1). Under this new problem setting, we aim to propose a theoretically guaranteed and empirically efficient safe RL algorithm that is applicable to real, large-scale problems.

## Related Work

**RL with (generalized) linear function approximation.** Function approximation, especially that based on deep neural networks (DNNs), has been one of the main reasons for the success of RL in recent years, and it has attracted more attention in both theory and practice [10, 32]. Given the difficulty of theoretical analysis of DNNs, linear MDPs [19, 22] characterized by linear transition kernels and reward are frequently used due to their tractability. The main interest of the theoretical analysis in existing studies has been the *efficiency* of RL algorithms. In fact, under the assumption of linear MDPs, recent studies (e.g., Jin et al. [18], Yang and Wang [39]) have proposed algorithms with a theoretical guarantee regarding sample efficiency. Wang et al. [38] then extended Jin et al. [18] to generalized linear model (GLM) settings and presented a practical and provably efficient algorithm under strictly weaker assumptions.

**Safe RL.** A constrained MDP (CMDP, [4]) is a popular model for formulating safe RL problems. In the CMDP formulation, an agent needs to maximize the expected cumulative reward while ensuring that the expected cumulative cost is less than a threshold. There has been a number of attempts to solve CMDPs [26], as represented by constrained policy optimization (CPO [2]), reward constrained policy optimization [33], Lagrangian-based actor-critic [7, 9, 13], primal-dual policy optimization [23, 39], and interior-point policy optimization [21]. As a recent, closely related work, Ding et al. [14] proposed an algorithm with theoretical guarantees in regard to regret and constraint violation for CMDPs with linear structures of transition, reward, and safety. In the previous papers mentioned above, however, a safety constraint is defined over a horizon; hence, they deal with less strict safety constraints than our paper that requires the agent to satisfy safety constraints *at every time step*. There has been research aimed at guaranteeing safety at every time step in both state-less and stateful settings. For state-less settings, SAFEOPT [31] and SAFE-LUCB [5] were respectively proposed for Bayesian optimization problems and linear stochastic bandit problems. For stateful settings, Turchetta et al. [34] proposed a GP-based SAFEMDP algorithm that extends SAFEOPT to stateful settings. Under the assumptions of the deterministic transition and regularity of safety, while Turchetta et al. [34] aimed to fully explore the safe region, Wachi et al. [37] proposed a GP-based algorithm for maximizing the cumulative reward under safety constraint.

## Our Contributions

We model a new class of safe exploration problems where features are locally available within a limited sensor range. This is a realistic formulation for some real-world robots with sensors, and it allows us to get rid of the regularity and Lipschitz assumptions that previous safe exploration

algorithms relied on. We develop safe policy optimization with a local feature (SPO-LF), an RL algorithm that maximizes the cumulative reward under safety constraints while predicting reward and safety function values on the basis of local features via GLMs. We analyze the theoretical properties of SPO-LF and prove that an agent will achieve a near-optimal policy with high probability while satisfying safety constraints at every time step. Finally, we evaluate our algorithm in two experiments. In the first experiment using Gym-MiniGrid [12], we show that our algorithm is 1) more efficient in terms of sample complexity and computational cost and 2) more applicable to large-scale problems than previous, highly-theoretical GP-based methods. In the second experiment using Safety-Gym [25], we show that our algorithm achieves comparable performance in terms of the reward to several notable safe RL baselines (e.g., CPO [2]) without executing even a single unsafe action.

## 2 Problem Formulation

We consider a safety-constrained MDP, defined as a tuple
$$\mathcal{M} = \langle \mathcal{S}, \mathcal{A}, f, H, r, g, \phi, \psi \rangle,$$
where $\mathcal{S}$ is a finite set of states $\{s\}$, $\mathcal{A}$ is a finite set of actions $\{a\}$, $f : \mathcal{S} \times \mathcal{A} \to \mathcal{S}$ is the deterministic state transition, and $H \in \mathbb{N}$ is the finite horizon. Also, $r : \mathcal{S} \to [0, 1]$ and $g : \mathcal{S} \to [0, 1]$ are bounded reward and safety functions. We assume that both $r$ and $g$ are *not known* a priori.[1] At every time step $t \in \mathbb{N}$, the agent must be in a *safe* state. Specifically, for a state $s_t$, the safety function value $g(s_t)$ must be above a threshold $h \in (0, 1]$; that is, the safety constraint is represented as $g(s_t) \geq h$.

In real applications, a robot is equipped with sensors and can infer the values of reward or safety functions by using such measurements as "hints." To formulate this mechanism, $\mathcal{M}$ contains two additional functions. First, $\phi : \mathcal{S} \to \mathbb{B}^d$ is an *a priori unknown* function for mapping a state $s$ to a $d$-dimensional feature vector, where $\mathbb{B}^d := \{x \in \mathbb{R}^d \mid \|x\| \leq 1\}$. For convenience, let $\phi_s := \phi(s)$ denote the feature vector for a state $s$. Second, we define a function $\psi : \mathcal{S} \to \mathcal{S}$ for returning a set of states within the sensor range from a certain state. Note that, while conventional methods for solving MDPs with linear function approximation assume the prior availability of the feature map for a whole state space, we consider the problem where features are locally obtained upon observation.

**Near-sighted and far-sighted observations.** On the basis of the two functions $\phi$ and $\psi$, we model two types of observation: *near-sighted observation* and *far-sighted observation*. Near-sighted observation is associated with direct observations of reward and safety, which is commonly used in typical RL settings including previous GP-based safe exploration studies. Using near-sighted observations, the agent obtains a set of environmental information that contains noisy observations of the reward and safety function values along with the feature vector for the current state. That is, the agent obtains $(s_t, \phi_{s_t}, y_r, y_g)$, where $y_r$ and $y_g$ are noisy observations of reward and safety, which are represented as $y_r = r(s) + n_t^r$ and $y_g = g(s) + n_t^g$, with independent zero-mean noises $n_t^r$ and $n_t^g$. Far-sighted observation is more related to feature vectors. In this observation scheme, an agent obtains features for $\psi(s)$. For convenience, let $\Psi_t \subseteq \mathcal{S}$ denote the set of states that have been observed at least once until time $t$; that is, $\Psi_t := \psi(s_0) \cup \psi(s_1) \cup \ldots \cup \psi(s_t)$. Intuitively, the agent needs to predict the reward and safety function values by improving the internal GLMs on the basis of near-sighted observations and applying them to the features obtained by far-sighted observations.

**Goal.** The quality of a policy $\pi : \mathcal{S} \to \mathcal{A}$ is evaluated in terms of the cumulative reward achieved by the agent under the safety constraint. The problem is formally written as

$$\text{maximize:} \quad V_{\mathcal{M}}^{\pi}(s_t) = \mathbb{E}\left[ \sum_{\tau=0}^{H} \gamma^{\tau} r(s_{t+\tau}) \,\middle|\, \pi \right] \quad \text{subject to:} \quad g(s_t) \geq h.$$

There are a number of previous studies on solutions to CMDPs, but most of them violate safety constraints during training (e.g., [2]). In contrast, we require the agent to learn a policy in $\mathcal{M}$ without even a single constraint violation. Without assumptions, it is intractable to solve the problem mentioned above. In the following, we list assumptions in this paper.

**Generalized linear function approximation.** The first assumption is on the structure of the reward and safety functions. In this paper, we are concerned with generalized linear models (GLMs) regarding the reward and safety functions that are characterized by a feature vector.

---

[1]For simplicity, we assume that the state transition function, $f$ is *known a priori*. Note that it is not difficult to extend our setting to a priori unknown $f$ by referring to Biyik et al. [8].

**Assumption 1.** *The reward and safety functions are respectively represented using GLMs that are characterized by unknown coefficient vectors $\boldsymbol{\theta}_r^*, \boldsymbol{\theta}_g^* \in \mathbb{R}^d$ and fixed, strictly increasing link functions $\mu_r, \mu_g : \mathbb{R} \to [0,1]$ such that*

$$r(\boldsymbol{s}) = \mu_r(\boldsymbol{\phi}_{\boldsymbol{s}}^\top \boldsymbol{\theta}_r^*) \quad and \quad g(\boldsymbol{s}) = \mu_g(\boldsymbol{\phi}_{\boldsymbol{s}}^\top \boldsymbol{\theta}_g^*).$$

Note that linear and logistic models are the special cases of GLM with $\mu(x) = x$ and $\mu(x) = \frac{1}{1+e^{-x}}$, respectively. To predict the coefficients, we obtain maximum likelihood estimators (MLEs) for reward and safety functions as in Filippi et al. [15]. Let $\tilde{\theta}_r$ and $\tilde{\theta}_g$ be the MLEs for reward and safety, respectively. Here, $\tilde{\theta}_r$ and $\tilde{\theta}_g$ are the unique solutions of the following equations.

$$\sum_{\tau=0}^{t} \left( r(\boldsymbol{s}_\tau) - \mu_r(\boldsymbol{\phi}_{\boldsymbol{s}_\tau}^\top \theta_r) \right) \boldsymbol{\phi}_{\boldsymbol{s}_\tau} = 0 \quad and \quad \sum_{\tau=0}^{t} \left( g(\boldsymbol{s}_\tau) - \mu_g(\boldsymbol{\phi}_{\boldsymbol{s}_\tau}^\top \theta_g) \right) \boldsymbol{\phi}_{\boldsymbol{s}_\tau} = 0.$$

The optimal solutions of the above equations can be found efficiently using such standard algorithms as Newton's method. We additionally make an assumption on the properties of the link functions.

**Assumption 2.** *Let $\diamond$ denote either $r$ or $g$. The link function $\mu_\diamond$ is twice differentiable, and the first and second order derivatives are respectively bounded by $L_\diamond$ and $M_\diamond$. Also, the link function $\mu_\diamond$ satisfies $\xi_\diamond = \inf_{\|\theta_\diamond - \theta_\diamond^*\| \le 1, \|\phi_{\boldsymbol{s}}\| \le 1} \dot{\mu}_\diamond(\boldsymbol{\phi}_{\boldsymbol{s}}^\top \theta_\diamond) > 0$.*

**Prior knowledge on safety.** Without prior information on the states known to be safe, an agent cannot guarantee safety at the first step. Hence, we make the following assumption.

**Assumption 3.** *The agent is initially in a set of states, $S_0 \subseteq \mathcal{S}$, that is known to be safe a priori.*

**Sub-Gaussian noise.** The final assumption is on the properties of the observation noise, which has been commonly made in previous work (e.g., Abbasi-yadkori et al. [1]).

**Assumption 4.** *Let $\diamond$ be either $r$ or $g$. The noise $n_t^\diamond$ is sub-Gaussian with fixed (positive) parameters $\sigma_\diamond$; that is, for all $t$, we have $\mathbb{E}\left[ e^{\omega_\diamond n_t^\diamond} \mid \mathcal{F}_{t-1}^\diamond \right] \le e^{\omega_\diamond^2 \sigma_\diamond^2 / 2}$, where $\{\mathcal{F}_n^\diamond\}$ is increasing sequences of sigma fields such that $n_t^\diamond$ is $\mathcal{F}_t^\diamond$-measurable with $\mathbb{E}\left[ n_t^\diamond \mid \mathcal{F}_{t-1}^\diamond \right] = 0$.*

## 3 Preliminary: Defining True Safe Space

In this section, we ask and describe what the true safe space is. Since the agent observes noisy measurements of reward and safety functions, it is impossible to know the exactly true reward and safety functions. This is particularly critical when dealing with safety; thus, we aim to conservatively estimate the safety function value up to a certain confidence $\epsilon \in \mathbb{R}_{\ge 0}$. As discussed in Turchetta et al. [34], to identify the safety of a state, we incorporate the safety constraint, reachability, and returnability. Under our settings, however, an agent is allowed to observe the feature vector for multiple neighboring states; hence, the existing definition of the true safe space cannot be directly used. Hereinafter, we explain how to represent the true safe space in $\mathcal{M}$.

First, we simply consider a set of states that satisfy the safety constraint with a margin of $\epsilon$. Given the set of states $X$, the following region can be identified as safe based on the far-sighted observation.

$$Y_\epsilon(X) = X \cup \{\boldsymbol{s} \in \mathcal{S} \mid \exists \boldsymbol{s}' \in X, \boldsymbol{s} \in \boldsymbol{\psi}(\boldsymbol{s}') : g(\boldsymbol{s}) - \epsilon \ge h\}.$$

Second, we define reachability. The agent may not be able to reach a state that is in $Y_\epsilon(\cdot)$ due to unsafe states in the middle of the pathway. Given a set $X$, the states in $\Psi$ that are reachable from $X$ in one step are given by $Y_{\text{reach}}(X) = X \cup \{\boldsymbol{s}' \in \Psi \mid \exists \boldsymbol{s} \in X, a \in \mathcal{A} : \boldsymbol{s}' = f(\boldsymbol{s}, a)\}$. Hence, an $n$-step reachable operator is given by $Y_{\text{reach}}^n(X) = Y_{\text{reach}}(Y_{\text{reach}}^{n-1}(X))$, where $Y_{\text{reach}}^1(X) = Y_{\text{reach}}(X)$. Finally, the set containing all the states that are reachable from $X$ along an arbitrary long path in $\Psi$ is defined as $\bar{Y}_{\text{reach}}(X) = \lim_{n \to \infty} Y_{\text{reach}}^n(X)$. Finally, even after arriving at a state with reachability, the agent may *not* have even a single option from among the viable actions for moving to another safe state. Hence, before moving to a state $\boldsymbol{s}$, we should consider whether or not there is at least one viable path from $\boldsymbol{s}$. The set of states from which the agent can return to a set $\bar{X}$ through another set of states $X$ in one step is given by $Y_{\text{return}}(X, \bar{X}) = \bar{X} \cup \{\boldsymbol{s} \in X \mid \exists a \in \mathcal{A} : f(\boldsymbol{s}, a) \in \bar{X}\}$. Thus, with $Y_{\text{return}}^1(X, \bar{X}) = Y_{\text{return}}(X, \bar{X})$, an $n$-step returnability operator is given by $Y_{\text{return}}^n(X, \bar{X}) = Y_{\text{return}}(X, Y_{\text{return}}^{n-1}(X, \bar{X}))$. The set containing all the states from which the agent can return to $\bar{X}$ along an arbitrary long path in $X$ is defined as $\bar{Y}_{\text{return}}(X, \bar{X}) = \lim_{n \to \infty} Y_{\text{return}}^n(X, \bar{X})$.

To guarantee safety, all the above requirements must be satisfied. As such, we define $Z(\cdot)$:

$$Z_\epsilon(X) = Y_\epsilon(X) \cap \bar{Y}_{\text{reach}}(X) \cap \bar{Y}_{\text{return}}(Y_\epsilon(X), X).$$

The safe set after repeating such expansions $n$-times is $Z_\epsilon^n(X) = Z_\epsilon(Z_\epsilon^{n-1}(X))$ with $Z_\epsilon^1(X) = Z_\epsilon(X)$. Finally, the true safe space, $\bar{Z}_\epsilon(X)$ is obtained by taking the limit in terms of $n$; that is, $\bar{Z}_\epsilon(X) = \lim_{n\to\infty} Z_\epsilon^n(X)$.

# 4 SPO-LF Algorithm

Our SPO-LF algorithm is outlined in Algorithm 1. On the basis of near-sighted and far-sighted observations, an agent first calculates the upper and lower bounds of reward and safety function values and predicts a safe region (lines $3-8$). The agent then optimizes its policy such that the cumulative reward is maximized while guaranteeing safety. For this purpose, the agent chooses the next state within the pessimistically identified safe space though this choice is basically based on the optimism in the face of uncertainty (OFU) principle in terms of reward (lines $9-11$). However, we observed a problem in that this pure approach often causes the agent to get stuck in a reward-poor state. As a work-around for this propose, we additionally propose an algorithm, called event-triggered safe expansion (ETSE) that encourages the agent to focus on the exploration of safety as necessary (lines $12-17$). Note that the proofs of the lemmas are in Appendix C.

---

**Algorithm 1** SPO-LF with ETSE

1: $C_0(\boldsymbol{s}) \leftarrow [h, \infty)$ for all $\boldsymbol{s} \in S_0$
2: **loop**
3: $\quad \Psi_t \leftarrow \Psi_{t-1} \cup O(\boldsymbol{s}_t)$
4: $\quad Q_t(\boldsymbol{s}) \leftarrow (1), C_t(\boldsymbol{s}) \leftarrow Q_t(\boldsymbol{s}) \cap C_{t-1}(\boldsymbol{s})$
5: $\quad l_t(\boldsymbol{s}) \leftarrow \min C_t(\boldsymbol{s}), u_t(\boldsymbol{s}) \leftarrow \max C_t(\boldsymbol{s})$
6: $\quad S_t^- \leftarrow \{\boldsymbol{s} \in \mathcal{S} \mid l_t(\boldsymbol{s}) \geq h\}$
7: $\quad S_t^+ \leftarrow \{\boldsymbol{s} \in \mathcal{S} \mid u_t(\boldsymbol{s}) \geq h\}$
8: $\quad \mathcal{X}_t^- \leftarrow (2), \mathcal{X}_t^+ \leftarrow (3)$
9: $\quad J^*(\boldsymbol{s}_t) \leftarrow (4)$
10: $\quad$ **if** $\arg\max_{\boldsymbol{s} \in Y_{\text{reach}}(\{\boldsymbol{s}_t\})} J^*(\boldsymbol{s}) \subseteq \mathcal{X}_t^-$ **then**
11: $\qquad \boldsymbol{s}_{t+1} \leftarrow \arg\max_{\boldsymbol{s} \in (\mathcal{X}_t^- \cap Y_{\text{reach}}(\{\boldsymbol{s}_t\}))} J^*(\boldsymbol{s})$
12: $\quad$ **else**
13: $\qquad \bar{J}^*(\boldsymbol{s}_t) \leftarrow (5)$
14: $\qquad$ **while** $\boldsymbol{s}_t \neq \arg\max_{\boldsymbol{s} \in \mathcal{X}_t^-} \bar{J}^*(\boldsymbol{s})$ **do**
15: $\qquad\quad \boldsymbol{s}_{t+1} \leftarrow \arg\max_{\boldsymbol{s} \in Y_{\text{reach}}(\{\boldsymbol{s}_t\})} \bar{J}^*(\boldsymbol{s})$
16: $\qquad$ **end while**
17: $\quad$ **end if**
18: **end loop**

---

## 4.1 Confidence Intervals

Since the reward and safety function values need to be "estimated" on the basis of the observations, we aim to achieve a provably safe and optimal RL algorithm by calculating their upper and lower bounds. Hence, we first derive theoretically guaranteed confidence intervals in terms of reward and safety. Since the accuracy of the predictions depends on the availability of the feature vector, we obtain the confidence bounds according to whether a state is inside or outside $\Psi_t$. Hereinafter, let the design matrix be $W_t = \sum_{\tau=1}^t \boldsymbol{\phi}_{\boldsymbol{s}_\tau} \boldsymbol{\phi}_{\boldsymbol{s}_\tau}^\top$. Also, the weighted $L_2$-norm of $\boldsymbol{\phi}$ associated with $W_t^{-1}$ is given by $\|\boldsymbol{\phi}\|_{W_t^{-1}} = (\boldsymbol{\phi}^\top W_t^{-1} \boldsymbol{\phi})^{1/2}$. Finally, the maximum and minimum singular values of a matrix are denoted as $\lambda_{\max}(\cdot)$ and $\lambda_{\min}(\cdot)$.

**Confidence intervals inside $\Psi_t$.** For all $\boldsymbol{s}$ in $\Psi_t$, the agent has an observed feature vector; hence, the reward and safety function values can be estimated more accurately by leveraging it. On the basis of Theorem 1 in Li et al. [20], we have the following lemma.

**Lemma 1.** *(Thm. 1 in [20]). Let $\delta_\diamond > 0$ be given and $\beta_\diamond = \frac{3L_\diamond \sigma_\diamond}{\xi_\diamond} \sqrt{\log(3/\delta_\diamond)}$. Assume $\lambda_{\min}(W_t) \geq 512\sigma_\diamond^2 M_\diamond^2 \xi_\diamond^{-4} \left(d^2 + \log \delta_\diamond^{-1}\right)$. Then, with a probability of at least $1 - \delta_\diamond$, the MLE satisfies*

$$| \diamond(\boldsymbol{s}) - \mu_\diamond(\boldsymbol{\phi}_{\boldsymbol{s}}^\top \tilde{\theta}_\diamond) | \leq \beta_\diamond \cdot \|\boldsymbol{\phi}_{\boldsymbol{s}}\|_{W_t^{-1}}.$$

The symbol $\diamond$ denotes either $r$ or $g$; that is, Lemma 1 respectively holds for reward and safety.

**Confidence intervals outside $\Psi_t$.** For $\boldsymbol{s}$ outside $\Psi_t$, the agent does not know even the feature vector. To obtain the confidence intervals outside $\Psi_t$, we should incorporate *any* possible feature vector; hence, confidence bounds are rather loose as presented below.

**Lemma 2.** *Let $\diamond$ denote either $r$ or $g$. Assume $\lambda_{\min}(W_t) \geq 512 \cdot \sigma_\diamond^2 M_\diamond^2 \xi_\diamond^{-4} \left(d^2 + \log \delta_\diamond^{-1}\right)$. Then, with a probability of at least $1 - \delta_\diamond$, for any $\boldsymbol{s} \in \mathcal{S} \setminus \Psi$, we have*

$$0 \leq \diamond(\boldsymbol{s}) \leq \mu_\diamond(\|\tilde{\theta}_\diamond\|) + \beta_\diamond \cdot \lambda_{\max}(W_t^{-1}).$$

The next lemma is on the relation between the upper bound defined above and the threshold, $h$.

**Lemma 3.** *Assume $\lambda_{\min}(W_t) \geq 512 \cdot \sigma_g^2 M_g^2 \xi_g^{-4} \left(d^2 + \log \delta_g^{-1}\right)$. Then, with a probability of at least $1 - \delta_g$, we have $\mu_g(\|\tilde{\theta}_g\|) + \beta_g \cdot \lambda_{\max}(W_t^{-1}) \geq h$.*

**Efficiency metric of exploration.** Because $\beta_r$ and $\beta_g$ are constants independent from $s$, the efficiency of the exploration of the reward and safety functions depends solely on $\|\phi\|_{W_t^{-1}}$ within $\Psi_t$ and $\lambda_{\max}(W_t^{-1})$ outside $\Psi_t$. Hence, useful actions for exploring the reward also contribute to the exploration of safety. This is an advantage in leveraging feature vectors in safe RL problems, and it leads to more efficient policy optimization compared with previous work such as Wachi and Sui [36] that requires the agent to explore the two functions separately.

**Assumption on $\lambda_{\min}(W_t)$.** As discussed in Li et al. [20], the assumption on $\lambda_{\min}(W_t)$ is satisfied under mild conditions. However, since we consider a safe exploration problem, it is extremely important to know how to ensure that the assumption on $\lambda_{\min}(W_t)$ is valid. This paper presents two methods; the first is to provide (typically a small number of) data for initializing GLMs, and the second is to prepare a sufficiently large $S_0$ and prohibit the agent from going outside of $S_0$, until the condition on $\lambda_{\min}(W_t)$ holds. In our experiment, we simply took the first approach.

## 4.2 Prediction of Safe Space

We now define two types of predicted safe space where safety is identified either optimistically or pessimistically, as discussed in Turchetta et al. [35]. First, we define the lower and upper bounds of safety. By Lemma 1 and 2, the confidence bounds on safety are represented as

$$Q_t(s) = \begin{cases} [\,\mu_g(\phi_s^\top \tilde{\theta}_g) \pm \beta_g \cdot \|\phi_s\|_{W_t^{-1}}\,] & \text{if} \quad s \in \Psi_t, \\ [\,0, \mu_g(\|\tilde{\theta}_g\|) + \beta_g \cdot \lambda_{\max}(W_t^{-1})\,] & \text{otherwise.} \end{cases} \tag{1}$$

We consider the intersection of $Q_t$ up to iteration $t$, which is defined as $C_t(s) = Q_t(s) \cap C_{t-1}(s)$, where $C_0(s) = [h, \infty]$ for all $s \in S_0$. The lower and upper bounds on $C_t(s)$ are denoted by $l_t(s) := \min C_t(s)$ and $u_t(s) := \max C_t(s)$, respectively.

**Predicted pessimistic safe space.** We now define the predicted *pessimistic* safe space. We first consider a set of states such that the safety constraint is satisfied with high probability. It is formulated using $l_t$; that is,

$$S_t^- := \{\, s \in \mathcal{S} \mid l_t(s) \geq h \,\}.$$

The desired *pessimistic* safe space, $\mathcal{X}_t^-$ is a subset of $S_t^-$ and also satisfies the reachability and returnability constraints; that is,

$$\mathcal{X}_t^- = S_t^- \cap \bar{Y}_{\text{reach}}(\mathcal{X}_{t-1}^-) \cap \bar{Y}_{\text{return}}(S_t^-, \mathcal{X}_{t-1}^-). \tag{2}$$

**Predicted optimistic safe space.** To optimize a policy, we need to incorporate the reward not only in $\mathcal{X}_t^-$ but also in a region that contains all the states that may turn out to be safe even with a small probability. We consider the set of states that may satisfy the safety constraint even with a small probability, which is represented as

$$S_t^+ := \{\, s \in \mathcal{S} \mid u_t(s) \geq h \,\}.$$

Note that, by Lemmas 2 and 3, for all states in $\mathcal{S} \setminus \Psi_t$, the upper and lower confidence bounds in terms of safety satisfy $l_t(s) < h \leq u_t(s)$. Hence, all states in $\mathcal{S} \setminus \Psi_t$ are identified as pessimistically unsafe but optimistically safe. Similarly to the pessimistic safe region, we again incorporate reachability and returnability. The resulting optimistic safe region is given by

$$\mathcal{X}_t^+ = S_t^+ \cap \bar{Y}_{\text{reach}}(\mathcal{X}_{t-1}^+) \cap \bar{Y}_{\text{return}}(S_t^+, \mathcal{X}_{t-1}^+). \tag{3}$$

## 4.3 Safe Policy Optimization based on OFU Principle

As discussed in Section 4.1, we can efficiently explore reward and safety simultaneously by focusing on $\|\phi\|_{W_t^{-1}}$ and $\lambda_{\max}(W_t^{-1})$. Hence, it turns out that we only need to balance exploration and

exploitation in terms of reward. For balancing 1) the exploration of reward and safety and 2) the exploitation of reward, we solve a Bellman equation on the basis of the OFU principle:

$$J^*(s_t) = \max_{s \in \mathcal{X}_t^+} \left[ R(s) + \gamma J^*(s) \right] \quad \text{where} \quad R(s) := \begin{cases} \mu_r(\phi_s^\top \tilde{\theta}_r) + \beta_r \|\phi_s\|_{W_t^{-1}} & \text{if } s \in \Psi_t, \\ \mu_r(\|\tilde{\theta}_r\|) + \beta_r \lambda_{\max}(W_t^{-1}) & \text{otherwise.} \end{cases} \tag{4}$$

Note that $R(s)$ represents the upper confidence bound with regard to reward. The next state to visit should be pessimistically safe and reachable from $s_t$ with one step; that is,

$$s_{t+1}^* = \underset{s \in (\mathcal{X}_t^- \cap Y_{\text{reach}}(\{s_t\}))}{\arg \max} J^*(s).$$

**Problem with this approach.** We observed that the above algorithm often causes the agent to get stuck in a state with a (relatively) small reward. The reason for this problem is as follows. When the state with the maximum $J^*$ is outside $\mathcal{X}_t^-$, the agent must execute the "second-best" action. The second-best solution is often the one where the agent tries to take the *stay* action and then visit $s_{t+1}^*$; then, the agent continues to stay at the current state.

**Event-triggered safe expansion.** To address the above problem, we propose an algorithm called event-triggered safe expansion (ETSE). We first define

$$\eta := \underset{s \in Y_{\text{reach}}(\{s_t\})}{\arg \max} J^*(s),$$

meaning the state that the agent will visit *if there is no safety constraint*. The ETSE algorithm is *triggered* only when $s_{t+1}^* \neq \eta$. Intuitively, if $\eta$ is within $\mathcal{X}_t^-$, the agent can simply execute the next optimal action to visit $s_{t+1}^*$; hence, the above problem arises only if $s_{t+1}^* \neq \eta$ is satisfied. This algorithm focuses on expanding the pessimistic safe region by visiting the states with a large uncertainty.

$$\bar{J}^*(s_t) = \max_{s_{t+1} \in \mathcal{X}_t^-} \left[ \|\phi_{s_{t+1}}\|_{W_t^{-1}} + \gamma \bar{J}^*(s_{t+1}) \right]. \tag{5}$$

On the basis of (5), the agent plans the trajectory so as to gain a lot of information, and the ETSE continues until the agent arrives at the state with the highest uncertainty. Once the ETSE finishes being applied, the agent aims to handle the exploration and exploitation dilemma by solving (4).

## 5 Theoretical Analysis

In this section, we present the theoretical results obtained for our SPO-LF. As a preliminary for the results, let us define the time step $t^*$, which can be used as an indicator that the agent has sufficiently understood the models of reward and safety. We now present the following lemma regarding $t^*$.

**Lemma 4.** *Let $C_1$ and $C_2$ be positive, universal constants. Also, let $t^*$ denote the smallest integer satisfying*

$$\lambda_{\min}(\Sigma)t - C_1\sqrt{td} - C_2\sqrt{t \log(\delta_g^{-1})} \geq \max\left\{ \frac{\beta_g}{\epsilon}, 1 \right\}$$

*with $\Sigma = \mathbb{E}[\phi_t \phi_t^\top]$. Then, for all $t > t^*$, the following inequalities hold.*

$$|\mu_g(\phi^\top \theta_g^*) - \mu_g(\phi^\top \tilde{\theta}_g)| \leq \epsilon \quad \text{and} \quad \sum_{\tau=t+1}^{t+H} \|\phi_{s_\tau}\|_{W_\tau^{-1}} \leq \sqrt{2Hd \log\left(\frac{t+H}{d}\right)}.$$

Finally, we present two main theorems. The first theorem is on the satisfaction of the safety constraint, and the second is on the near-optimality of the acquired policy.

**Theorem 1.** *Assume that the noise of safety is $\sigma_g$-sub-Gaussian and $\lambda_{\min}(W_t) \geq 512 \cdot \sigma_g^2 M_g^2 \xi_g^{-4} \left( d^2 + \log \delta_g^{-1} \right)$ holds. Suppose that the next state is chosen from $\mathcal{X}_t^-$. Then, the following statement holds with a probability of at least $1 - \delta_g$:*

$$\forall t \geq 0, \quad g(s_t) \geq h.$$

The proof is in Appendix E. The agent chooses the next state to visit within $\mathcal{X}_t^-$ ($\subseteq S_t^-$), and $S_t^-$ is characterized with $l(s)$. Since $g(s)$ is greater than $l(s)$ with a probability of at least $1 - \delta_g$, our algorithm guarantees that the agent will execute only safe actions.[2]

---

[2]Unlike Turchetta et al. [34] or Wachi and Sui [36], this paper does not provide a theory on the completeness (i.e., convergence to the true safe region) of the predicted safe region. This is because our algorithm explores safety as a result of balancing exploration and exploitation for reward, which contributes to its sample-efficiency.

**Theorem 2.** *Set $T^* = t^* + |\bar{Z}_0(S_0)|$ and*

$$\epsilon_V = V_{\max} \cdot \delta_g + \beta_r(\delta_r)\sqrt{2Hd\log\left(\frac{t+H}{d}\right)},$$

*where $V_{\max}$ is the upper bound of the value function. Assume that the noise of reward and safety is sub-Gaussian and $H > T^*$. Then, with high probability,*

$$V^{\pi_t}(\boldsymbol{s}_t) \geq V^*(\boldsymbol{s}_t) - \varepsilon_V,$$

*— i.e., the algorithm is $\varepsilon_V$-close to the optimal policy — for all but $T^*$ time steps while guaranteeing safety with a probability of at least $1 - \delta_g$.*

The proof is in Appendix E. An outline of the proof is as follows. At each iteration, the agent encounters one out of the following two events. In one event, the ETSE algorithm is not triggered, and the agent can execute an action to purely optimize its policy. In the other event, $\boldsymbol{s}^*_{t+1} \neq \eta$ is satisfied, and the agent focuses on exploring the safe regions. The worst case scenario for the sample efficiency is when the second event continues to occur; that is, the agent is required to explore the whole safe space. However, after a sufficiently large number of time steps greater than $T^*$, the agent can achieve an $\varepsilon_V$-close policy by deciding the next action on the basis of the OFU principle in a fully-explored safe region. Note that, in the case of a short horizon, the agent cannot achieve the near-optimal policy; hence, we assume $H > T^*$.

## 6   Experiments

We evaluate the performance of our SPO-LF in two experiments. In the first experiment using Gym-MiniGrid [12], we show that our algorithm is better than existing ones backed by theory including [36] in terms of sample efficiency and scalability. In the second experiment based on Safety-Gym [25], we compare our algorithm with advanced deep RL algorithms incorporating constraints. We also demonstrate the effectiveness of the ETSE algorithm. For future research, our code is open-sourced.[3]

### 6.1   Grid World

**Settings.**   We constructed a simulation environment based on Gym-MiniGrid [12]. We considered a $25 \times 25$ grid in which each grid was associated with a randomly generated feature vector with the dimension $d = 5$. Coefficients for reward and safety were also randomly generated; thus, true reward and safety function values were allocated to each grid along with a feature vector. At every time step, an agent chose the next action from five candidates: *stay, up, right, down*, and *left*. The agent predicted the reward and safety functions while learning the relationship between the feature vector and the two functions with GLMs. In this simulation, we allowed the agent to 1) observe the feature vector in the $7 \times 7$ neighboring grids that are in the front of the agent and 2) obtain the feature vector and noisy measurements of the reward and safety values at the current state. As prior information, the agent received 10 samples of $(\boldsymbol{\phi}, g)$, to initialize the internal GLM model regarding safety. We considered a linear model with $\mu(x) = x$ for both reward and safety. Finally, we set $\gamma = 0.999$, $\delta_r = \delta_g = 0.05$, and $h = 0.1$, and optimized a policy with policy iteration.

We compared our SPO-LF with the following six baselines. The first is called ORACLE, and it has a non-exploratory agent who knows the true reward and safety functions a priori. The second is called UNSAFE GLM, in which a safety-agnostic agent maximizes a cumulative reward on the basis of GLMs. The third is called RANDOM, in which an agent randomly chooses the next action. The last three baselines are all stepwise approaches as in Wachi and Sui [36]. The fourth baseline is STEP SAFE GLM, in which an agent first expands the safe region and then optimizes the cumulative reward in the certified safe region while learning the reward and safety function structures via GLMs. The fifth is GP (FEATURE), which is a GP-based stepwise safe exploration and optimization where the inputs of GP are $d$-dimensional feature vectors. The final baseline is called GP (STATE), a stepwise approach proposed in Wachi and Sui [36] under the assumption that the reward and safety functions have regularity with regard to state. We used the reward and the number of unsafe actions as evaluation metrics.

---

[3]`https://github.com/akifumi-wachi-4/spolf`

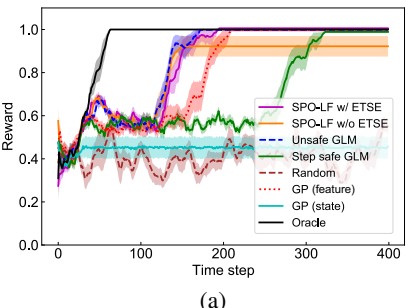 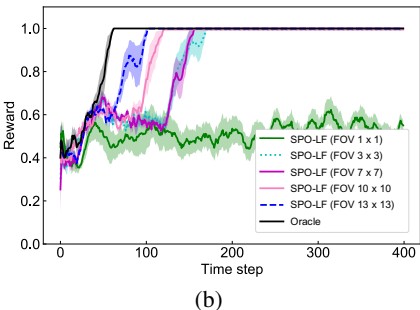

(a)               (b)

Figure 2: (a) Comparison of performance of SPO-LF and baselines. Standard error is shown as shaded area. (b) Comparison of performance of the SPO-LF with different sizes of field of view.

**Results.** Figure 2(a) compares the performance with regard to the cumulative reward of each algorithm. SPO-LF, UNSAFE GLM, STEP SAFE GLM, and GP (FEATURE) achieved a near-optimal policy. The convergence of SPO-LF was faster than STEP SAFE GLM and GP (FEATURE) but slower than UNSAFE GLM. This is because SPO-LF could not always execute the optimal action for maximizing the cumulative reward to guarantee safety. Also, without ETSE, the agent often got stuck in a state with a non-optimal reward. Figure 2(b) compares the performance of SPO-LF with different sizes of field of view (FOV) from $1 \times 1$ to $13 \times 13$. For a larger FOV, the convergence of the algorithm got faster. Given that SPO-LF (FOV $1 \times 1$) corresponding to previous work (e.g., [34, 36]) dealt with only near-sighted observations, it is useful to incorporate

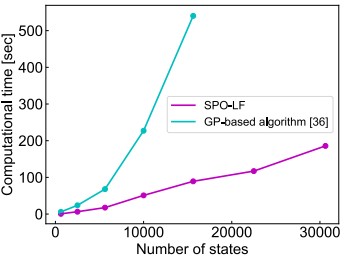

Figure 3: Computational time of GLM-based SPO-LF and GP-based algorithm (i.e., Wachi and Sui [36]).

far-sighted observations as in our new framework. Figure 3 compares the computational time for different environment sizes. SPO-LF is more scalable and applicable to large-scale problems given that the GP-based algorithms did not run in a reasonable amount of time for large environments such as that with $150 \times 150$ states. In the GLM-based methods, most of the computational cost when obtaining confidence bounds is for calculating $\lambda_{\max}(W_t^{-1})$. This computational complexity is $\mathcal{O}(d^3)$, which does neither depend on the number of states nor samples. Finally, *the average number of unsafe actions* was **3.4** for UNSAFE GLM, **49.0** for RANDOM, and **0.0** for the other methods.

### 6.2 Safety-Gym

**Settings.** The second experiment was based on Safety-Gym [25] with continuous state and action spaces. The objective of this experiment was to compare our SPO-LF with several notable safe RL algorithms including CPO [2], PPO-Lagrangian, and TRPO-Lagrangian. The last two baselines are ones that introduce that Lagrangian method into a safety-agnostic algorithm (i.e., PPO [29] and TRPO [28]) to incorporate constraints (for more details, see Section 5 in Ray et al. [25]). In this experiment, as shown in Figure 4(a), one goal (in green) and multiple hazards (in blue) were randomly placed, and the agent (in red) needed to reach the goal without visiting hazards. For encouraging the agent to reach the goal, we set two kinds of reward: 1) a reward of $1.0$ when arriving at the goal and 2) a reward of $0.01 \cdot \Delta_{\text{goal}}$ when getting closer to the goal, where $\Delta_{\text{goal}}$ is the amount of decrease in distance from the goal. To implement SPO-LF, we discretized the environment into $50 \times 50$ grids and used a computationally inexpensive policy iteration algorithm proposed in Scherrer et al. [27] for policy optimization. The feature vector $\phi$ was constructed by LiDAR observations (i.e., the distance from the goal or hazards). Since reward and safety function values vary steeply according to the existence of the goal or hazards, we use the sigmoid function $\mu(x) = \frac{1}{1+e^{-x}}$ as the link function. As for implementing the three baselines, our experiments largely depended on the "safety-starter-agent" repository (`https://github.com/openai/safety-starter-agents.git`). For

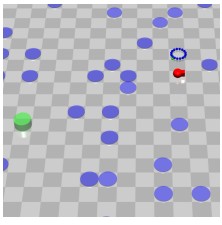
(a) Environment

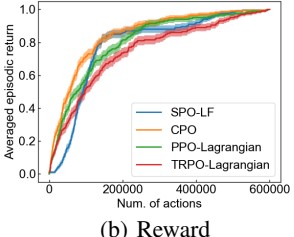
(b) Reward

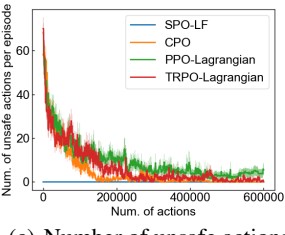
(c) Number of unsafe actions

Figure 4: Experiment with Safety-Gym. (a) Example screen capture from our simulation environment with a goal and hazards. (b) Average reward over episodes. (c) Number of unsafe actions per episode. Note that our SPO-LF did not execute even a single unsafe action.

the initialization, we provided all the agents with 40 reward and safety samples as prior information. For more detailed experimental settings, see Appendix F.

**Results.** Figures 4(b) and 4(c) show the experimental results. We used the reward and the number of unsafe actions as evaluation metrics and calculated the mean and standard error after running each method 200 times. As shown in Figure 4(b), our SPO-LF exhibited comparable performance to the baselines with regard to the sample efficiency. However, our algorithm *did not execute even a single unsafe action*, while the other three methods optimized their policies while making mistakes.

## 7  Conclusion

We formulated a new problem characterized by safety-constrained MDPs with local feature and then proposed the SPO-LF algorithm for safely optimizing a policy in an a priori unknown environment. SPO-LF efficiently and safely optimizes a policy by leveraging feature information while expanding the safe space as necessary by using the ETSE algorithm. Theoretical analysis showed that our algorithm obtains a near-optimal policy while guaranteeing safety, with a high probability. Our experiments showed that our algorithm 1) achieved better efficiency and scalability than previous safe exploration methods with theoretical guarantees and 2) behaved more safely than existing advanced deep RL methods with constraints.

We consider that our proposed method compensates for the shortcomings of safe RL methods with theoretical guarantees (e.g., [34], [36]) and advanced deep RL methods (e.g., [2]), which potentially sets out a research direction to bridge the gap between two distinct methods. However, there are several neglected problems we as the community should address in future work. One of the biggest problems is that our proposed algorithm (that is more scalable than highly-theoretical ones) is still far from practical in real applications with continuous state and action spaces. It would be an interesting direction to develop as scalable algorithm as advanced deep RL algorithms while maintaining theoretical safety guarantee.

Finally, we believe that safety is an essential requirement for applying RL in many real problems and have not found any negative societal impact of our algorithm. However, we need to remain aware that any sequential decision-making algorithms are vulnerable to misuse and ours is no exception.

## Acknowledgments

We deeply appreciate the anonymous reviewers for constructive comments. This work is partially funded by Tsinghua GuoQiang Research Institute.

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
