# Appendices

## A. Summary of Upper and Lower Bounds.

For easy understanding, we summarize the upper and lower bounds in terms of reward and safety, inferred by GLMs.

Table 1: Confidence bounds of reward and safety functions.

| | $s \in \Psi_t$ (FEATURE AVAILABLE) | $s \notin \Psi_t$ (FEATURE UNAVAILABLE) |
|---|---|---|
| REWARD | $[\,\mu(\phi_s^\top \tilde{\theta}_r) \pm \beta_r \cdot \|\phi_s\|_{W_t^{-1}}\,]$ | $[\,0, \mu(\|\tilde{\theta}_r\|) \pm \beta_r \cdot \lambda_{\max}(W_t^{-1})\,]$ |
| SAFETY | $[\,\mu(\phi_s^\top \tilde{\theta}_g) \pm \beta_g \cdot \|\phi_s\|_{W_t^{-1}}\,]$ | $[\,0, \mu(\|\tilde{\theta}_g\|) + \beta_g \cdot \lambda_{\max}(W_t^{-1})\,]$ |

## B. Preliminary Lemma

**Lemma 5.** *For two arbitrary functions $f_1(x)$ and $f_2(x)$, the following inequality holds:*

$$\max_x f_1(x) - \max_x f_2(x) \geq \min_x (f_1(x) - f_2(x)).$$

*Proof.* For two arbitrary functions $f_1(x)$ and $f_2(x)$, the following inequalities hold:

$$\max_x \{f_2(x) - f_1(x)\} + \max_x f_1(x) \geq \max_x f_2$$
$$\max_x f_1(x) - \max_x f_2(x) \geq - \max_x \{f_1(x) - f_2(x)\}$$
$$= \min_x \{f_1(x) - f_2(x)\}.$$

Then, the desired lemma is obtained. $\square$

## C. Lemmas on GLMs

### C.0. Preliminary lemma on GLMs

**Lemma 6.** *Let $\theta_g^*$ be the true coefficient for safety. Then, we have $\mu_g(\|\theta_g^*\|) \geq h$.*

*Proof.* By Cauchy–Schwarz inequality, for all $\phi_s$ such that $\|\phi_s\| \leq 1$, the following inequality holds:

$$\|\theta_g^*\| \geq \phi_s^\top \theta_g^*.$$

By Assumption 1, $\mu_g(\cdot)$ is a strictly increasing function; that is,

$$\mu_g(\|\theta_g^*\|) \geq \mu_g(\phi_s^\top \theta_g^*).$$

Assumption 3 implies that there exists $\phi_s$ such that

$$\mu_g(\phi_s^\top \theta_g^*) \geq h.$$

Therefore, we obtained the desired lemma. $\square$

### C.1. Proofs of Lemma 1

*Proof.* See Theorem 1 in Li et al. [20]. $\square$

## C.2. Proof of Lemma 2

*Proof.* First, we obtain the upper bound of $\diamond(s)$ for any $s \in \mathcal{S}$. Let $\phi^*_{\diamond,s}$ be the feature for achieving the maximum value of $\diamond$. Then, the following chain of equations hold, with a probability of at least $1 - \delta_\diamond$:

$$
\begin{aligned}
\max_{s \in \mathcal{S}} \diamond(s) &= \mu_\diamond(\langle \phi^*_{\diamond,s}, \theta^*_\diamond \rangle) \\
&\leq \mu_\diamond(\langle \phi^*_{\diamond,s}, \tilde{\theta}_\diamond \rangle) + \beta_\diamond \cdot \|\phi^*_{\diamond,s}\|_{W_n^{-1}} \\
&\leq \mu_\diamond(\|\tilde{\theta}_\diamond\|) + \beta_\diamond \cdot \|\phi^*_{\diamond,s}\|_{W_n^{-1}} \\
&\leq \mu_\diamond(\|\tilde{\theta}_\diamond\|) + \beta_\diamond \cdot \lambda_{\max}(W_n^{-1}).
\end{aligned}
$$

In the above chain of inequalities, we used Cauchy–Schwarz inequality and $\|\phi^*_{\diamond,s}\| \leq 1$ in the second line and then used $\|\phi^*_{\diamond,s}\|_{W_n^{-1}} \leq \lambda_{\max}(W_n^{-1})$ in the third line.

The lower bound can be simply obtained by definition; that is,

$$
\min_{s \in \mathcal{S}} \diamond(s) \geq 0.
$$

Then, we have the desired lemma. $\qquad\square$

## C.3. Proof of Lemma 3

*Proof.* By Lemma 2, with a probability of at least $1 - \delta_g$, we have the following inequality in terms of the confidence bounds of the safety function:

$$
0 \leq \max_{s \in \mathcal{S}} g(s) \leq \mu_g(\|\tilde{\theta}_g\|) + \beta_g \cdot \lambda_{\max}(W_n^{-1}).
$$

Also, by combining Lemma 6 and

$$
\mu_g(\|\theta^*_g\|) \leq \mu_g(\|\tilde{\theta}_g\|) + \beta_g \cdot \lambda_{\max}(W_n^{-1}),
$$

we have

$$
h \leq \mu_g(\|\theta^*_g\|) \leq \mu_g(\|\tilde{\theta}_g\|) + \beta_g \cdot \lambda_{\max}(W_n^{-1}).
$$

Finally, we obtained the desired lemma. $\qquad\square$

## C.4. Proof of Lemma 4

*Proof.* For all $t > t^*$, the following chain of inequalities hold:

$$
\begin{aligned}
\min\left\{\frac{\epsilon}{\beta_g}, 1\right\} &\geq \left(\lambda_{\min}(\Sigma)t - C_1\sqrt{td} - C_2\sqrt{t\log(\delta_g^{-1})}\right)^{-1} \\
&\geq \lambda_{\max}(W_n^{-1}).
\end{aligned}
$$

In the above calculation, we used Lemma 1 in Li et al. [20] and $\lambda_{\max}(W_n^{-1}) = \frac{1}{\lambda_{\min}(W_n)}$. Therefore, the following two statements are satisfied. First, because we have $\beta_g\lambda_{\max}(W_n^{-1}) \leq \epsilon$, the following inequalities hold:

$$
\begin{aligned}
|\mu_g(\phi^\top\theta^*_g) - \mu_g(\phi^\top\tilde{\theta}_g)| &\leq \beta_g\lambda_{\max}(W_n^{-1}) \\
&\leq \epsilon.
\end{aligned}
$$

Second, by Lemma 2 in Li et al. [20], the following inequality holds under the condition that $\lambda_{\max}(W_n^{-1}) \leq 1$:

$$
\sum_{\tau=t+1}^{t+H} \|\phi_{s_\tau}\|_{W_\tau^{-1}} \leq \sqrt{2Hd\log\left(\frac{t+H}{d}\right)}.
$$

$\qquad\square$

## D. Lemmas on Near-optimality

**Lemma 7.** *Let $J^*(s_t)$ be the value function calculated by our algorithm. Then, $J^*(s_t)$ satisfies the following inequality:*

$$J^*(s_t) \geq V^*(s_t).$$

*Proof.* Let $I_Z : \mathcal{S} \to \{0, 1\}$ denote the following safety indicator function:

$$I_Z(s) := \begin{cases} 1 & \text{if } s \in \bar{Z}_{\epsilon_g}(S_0), \\ 0 & \text{otherwise.} \end{cases} \tag{6}$$

Then, the following chain of equations and inequalities holds:

$$
\begin{aligned}
& J^*(s_t) - V^*(s_t) \\
& = \max_{s_{t+1} \in \mathcal{X}_t^+} \left[\, R(s_{t+1}) + \gamma J^*(s_{t+1}) \,\right] - \max_{s_{t+1} \in \bar{Z}_{\epsilon_g}(S_0)} \left[\, r(s_{t+1}) + \gamma V_{\mathcal{M}}^*(s_{t+1}) \,\right] \\
& \geq \max_{s_{t+1} \in \bar{Z}_{\epsilon_g}(S_0)} \left[\, R(s_{t+1}) + \gamma J^*(s_{t+1}) \,\right] - \max_{s_{t+1} \in \bar{Z}_{\epsilon_g}(S_0)} \left[\, r(s_{t+1}) + \gamma V_{\mathcal{M}}^*(s_{t+1}) \,\right] \\
& = \max_{s_{t+1}} \left[\, I_Z(s_{t+1})\{R(s_{t+1}) + \gamma J^*(s_{t+1})\} \,\right] - \max_{s_{t+1}} \left[\, I_Z(s_{t+1})\{r(s_{t+1}) + \gamma V_{\mathcal{M}}^*(s_{t+1})\} \,\right] \\
& \geq \min_{s_{t+1}} \left[\, I_Z(s_{t+1})\{R(s_{t+1}) - r(s_{t+1})\} + \gamma I_Z(s_{t+1})\{J^*(s_{t+1}) - V^*(s_{t+1})\} \,\right].
\end{aligned}
$$

The second line follows from $\mathcal{X}_t^+ \supseteq \bar{Z}_{\epsilon_g}(S_0)$, and the third line follows from the definition of $I_Z$. Also, the fourth line follows from Lemma 5. By definition of $R(s)$, the following equation holds with a probability of at least $1 - \delta^r$:

$$\min_{s_t} [\, J^*(s_t) - V^*(s_t) \,] \geq \gamma \cdot \min_{s_{t+1}} [\, I_Z(s_{t+1})\{J^*(s_{t+1}) - V^*(s_{t+1})\} \,]$$

Repeatedly applying this equation proves the desired lemma. Therefore, we have

$$J^*(s_t) \geq V^*(s_t)$$

with high probability. $\qquad \square$

**Lemma 8. (Generalized induced inequality)** *Let $\phi, r, g$ and $\hat{\phi}, \hat{r}, \hat{g}$ be the feature and reward function (including the exploration bonus) and safety function that are identical on some set of states $\Omega$ — i.e., $\phi = \hat{\phi}$, $r = \hat{r}$, and $g = \hat{g}$ for all $s \in \Omega$. Let $P(A_\Omega)$ be the probability that a state not in $\Omega$ is generated when starting from state $s$ and following a policy $\pi$. Assume value is bound in $[0, V_{\max}]$, then*

$$V^\pi(r, s, \phi, g) \geq V^\pi(\hat{r}, s, \hat{\phi}, \hat{g}) - V_{\max} P(A_\Omega),$$

*where we now make explicit the dependence of the value function on the reward.*

*Proof.* The lemma follows from Lemma 8 in Strehl and Littman [30]. $\qquad \square$

**Lemma 9.** *Suppose that the agent optimizes the policy based on our* SPO-LF*. After $T^*$ time step, let $A_\Omega$ be the event in which the agent escapes from $\Omega$. Then, for all $t > T^*$, we have*

$$P(A_{\mathcal{X}_t^-}) \leq \delta_g.$$

*Proof.* Hereinafter, let $\bar{q}_t$ be the sequence of states planed by the policy $\pi_t$ obtained by the algorithm at time $t$: that is,

$$\bar{q}_t := \{\, \bar{s}_t, \bar{s}_{t+1}, \dots, \bar{s}_{t+H} \,\},$$

where $\bar{s}_t = s_t$ and $\bar{s}_{t+\tau+1} = f(\bar{s}_{t+\tau}, \pi_t(\bar{s}_{t+\tau}))$ for all $\tau \in [0, H-1]$.

First, let us consider the case that prediction of the safety function values are collect, which occurs with a probability of at least $1 - \delta_g$ (we call EVENT A). While the agent optimizes its policy, one of the following two events happens.

- EVENT A-1: $\bar{q}_t \subseteq \mathcal{X}_t^-$.

- EVENT A-2: $\bar{q}_t \nsubseteq \mathcal{X}_t^-$.

When EVENT A-1 occurs, the optimal policy calculated based on the current information can be obtained only in $\mathcal{X}_t^-$. Hence, the agent will not go outside $\mathcal{X}_t^-$ as far as it follows the current policy. The worse-case for the agent is when EVENT A-2 continues to occur and the ETSE algorithm is triggered at every time step. Even under this worst condition, however, after $t^*$ time steps, the confidence interval on safety is bounded by $\epsilon$ and then the agent can identify whether or not a state is safe immediately on the basis of far-sighted observations. After the agent learns the safety function model up to $\epsilon$-accuracy, it takes at most $T^* - t^* = |R_0(S_0)|$ time steps until $\mathcal{X}_t^-$ is sufficiently expanded.

Second, if the prediction of safety is wrong, we cannot avoid the situation that the agent escapes from $\mathcal{X}_t^-$, which occurs with a probability of at most $\delta_g$.

In summary, for all $t > T^*$, only if the prediction of safety is inaccurate, the agent may go outside of $\mathcal{X}_t^-$. This event (i.e., Event B) happens with a (small) probability at most $\delta_g$. Hence, the desired lemma is now proved. $\qquad\square$

## E. Main Theorems

### E.1. Proof of Theorem 1

*Proof.* By Lemma 1 and Lemma 3, for all $s \in \mathcal{S}$, with a probability of at least $1 - \delta_g$, we have
$$g(s) \geq l(s).$$
Our SPO-LF requires the agent to visit only the state satisfying $l(s) \geq h$; hence, the safety constraint
$$g(s) \geq h$$
is satisfied with a probability of at least $1 - \delta_g$. $\qquad\square$

### E.2. Proof of Theorem 2

*Proof.* Define $\tilde{r}$ and $\tilde{g}$ as the reward function (with the exploration bonus) and safety function, which are used by the SPO-LF. Let $\hat{r}$ be a reward function equal to $r$ on $\Omega$ and equal to $\tilde{r}$ elsewhere. Furthermore, let $\tilde{\pi}$ be the policy followed by the SPO-LF at time $t$, that is, the policy calculated on the basis of the feature $\phi$ and predicted reward $\tilde{r}$ and safety $\tilde{g}$ inferred by the predicted coefficients, (i.e., $\tilde{\theta}_r$ and $\tilde{\theta}_g$). Finally, let $A_\Omega$ be the event in which $\tilde{\pi}$ escapes from $\Omega$. Then, by Lemma 8, we have
$$V^{\pi_t}(r, s_t, \phi_{s_t}, g) \geq V^{\tilde{\pi}}(\hat{r}, s_t, \phi_{s_t}, g) - V_{\max} P(A_\Omega).$$
In addition, note that, for all $t \geq t^*$ and $s_t \in \mathcal{X}_t^-$, because $\hat{r}$ and $\tilde{r}$ differ by at most $\beta_g \|\phi_{s_t}\|_{W^{-1}}$ at each state,

$$
\begin{aligned}
|V^{\tilde{\pi}}(\hat{r}, s_t, \phi_{s_t}, \tilde{g}) - V^{\tilde{\pi}}(\tilde{r}, s_t, \phi_{s_t}, \tilde{g})| &\leq \sum_{\tau=1}^{H} \beta_r \|\phi_{s_{t+\tau}}\|_{W^{-1}} \\
&= \beta_r \cdot \sum_{\tau=1}^{H} \|\phi_{s_{t+\tau}}\|_{W^{-1}} \\
&\leq \beta_r \cdot \sqrt{2Hd \log\left((t+H)/d\right)}. \quad (7)
\end{aligned}
$$

Here, consider the case of $\Omega = \mathcal{X}_{t^*}^-$. Once the safe region is fully explored, $P(A_{\mathcal{X}_{t^*}^-}) \leq \delta_g$ holds after $T^*$ time steps. Then, the following chain of equations and inequalities holds:

$$
\begin{aligned}
V^{\pi_t}(r, s_t, \phi_{s_t}, g) &\geq V^{\tilde{\pi}}(\hat{r}, s_t, \phi_{s_t}, g) - V_{\max} \cdot P(A_\Omega) \\
&= V^{\tilde{\pi}}(\hat{r}, s_t, \phi_{s_t}, g) - V_{\max} \cdot P(A_{\mathcal{X}^-}) \\
&\geq V^{\tilde{\pi}}(\hat{r}, s_t, \phi_{s_t}, g) - V_{\max} \cdot \delta_g \\
&\geq V^{\tilde{\pi}}(\tilde{r}, s_t, \phi_{s_t}, g) - V_{\max} \cdot \delta_g - \beta_r \cdot \sqrt{2Hd \log\left((t+H)/d\right)} \\
&= J_{\mathcal{X}}^*(\tilde{r}, s_t, \phi_{s_t}, g) - V_{\max} \cdot \delta_g - \beta_r \cdot \sqrt{2Hd \log\left((t+H)/d\right)} \\
&\geq V^*(r, s_t, \phi_{s_t}, g) - V_{\max} \cdot \delta_g - \beta_r \cdot \sqrt{2Hd \log\left((t+H)/d\right)}
\end{aligned}
$$

In this derivation, the second line follows from the assumption of $\Omega = \mathcal{X}^-$, the third line follows from $P(A_{\mathcal{X}^-}) \leq \delta_g$, the fourth line follows from (7), the fifth line follows from the fact that $\tilde{\pi}$ is precisely the optimal policy for $\tilde{R}$ and $b$, the sixth line follows from Lemma 7, and the final line follows from the definition of $\epsilon_V$. $\qquad\square$

## F. Additional Information on Experiments

Note that the source code we used for our experiment is in the supplemental material. Also, a video to illustrate how our SPO-LF works is in the supplementary material.

### F.1. Grid World

The source code for this experiment is contained in "grid_world" directory. In this experiment, we used Intel(R) Xeon(R) Gold 6248 CPU with 512 GB RAM. The number of iteration was 400. We used the following parameters as mentioned in the main paper, which are same throughout every method: $\gamma = 0.999$, $\delta_r = 0.05$, $\delta_g = 0.05$, $h = 0.1$, $d = 5$. For getting the results in Figure 2, we ran the simulation 100 times and calculated the mean and standard error for every method. When obtaining the Figure 3, due to the computational cost, we reduced $\gamma$ and number of simulation to $0.98$ and $5$, respectively. Example screen shot of our simulation environment is shown in Figure 5.

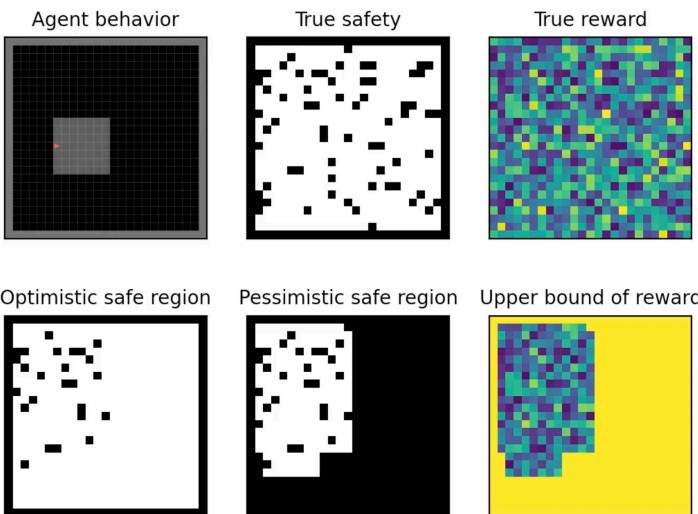

Figure 5: Example screenshot of our grid world experiment.

### F.2. Safety-Gym

We present the detailed information regarding our experiments using Safety-Gym [25]. In this experiment, we used Intel(R) Xeon(R) Platinum 8180 CPU with 1 TB RAM. We conducted an experiment using Safety-Gym environment with a goal and multiple hazards. The objective of the agent is to reach the goal without visiting hazards using a LiDAR sensor with 16 bins around a full circle for sensing.

For fair comparison, we used as many same parameters as possible for every method. The followings are the examples of the common parameters.

- Number of (different) environments for Monte-Carlo simulation: 200
- Discount factor: $\gamma = 0.99$
- Number of samples for initialization: 40

Reward and safety samples for initialization were created by letting agents observe "test environment" with only one goal and one hazard. This (small number of) samples are particularly significant for

our proposed method to initialize the GLM and enable the agent to guarantee safety. Note that, for fair comparison, we also provided prior information for other methods.

**Implementation of CPO, PPO-Lagrangian, and TRPO-Lagrangian.** Our baseline implementations are heavily dependent on the *safety-starter-agent* repository (`https://github.com/openai/safety-starter-agents.git`) as mentioned in the main paper. We used the following parameters.

- Number of epochs: 600
- Max. length of episode: 1000
- Optimizer: Adam
- hidden size: (64, 64)

**Implementation of** `SPO-LF`. Safety-Gym is a benchmark originally developed for deep RL algorithms. Unlike the three baselines, our algorithm is basically based on dynamic programming methods such as policy iteration to solve Bellman equations. Hence, when we test our proposed method, we discretized the state space into $50 \times 50 = 2,500$ grids and the action space into five (i.e., go north, go east, go south, go west, and stay). We converted discrete actions to a series of continuous actions to navigate the agent in Safety-Gym environment.

We constructed the feature vector using LiDAR observations for the goal and hazards. Specifically, the feature vector concatenate such observations and bias constant of 1, which is represented as

$$\phi = [\, o_{\text{goal}}, o_{\text{hazard}}, 1 \,],$$

where $o_{\text{goal}}$ and $o_{\text{hazard}}$ are respectively the LiDAR observations for the closest goal and hazards. We used the following parameters in our experiment.

- Algorithm to solve Bellman equation: Approximate Modified Policy Iteration [27]
- Link function: sigmoid function
- Safety threshold: $h = 0.7$