# OpenReview forum: "Safe Policy Optimization with Local Generalized Linear Function Approximations"
_NeurIPS.cc/2021/Conference — NeurIPS 2021 Poster_

### Official Review · Reviewer_E6K1 · 2021-07-15

**Rating:** 8
**Confidence:** 4

**Summary:**

This paper studies the problem of safe exploration in reinforcement learning contexts. It develops a new algorithm which, by making several assumptions, can stay safe while exploring and achieves good sample efficiency. The proposed approach incorporates sensor measurements ("far-sighted observations") and estimates of safety based on those observations, as well as the idea of an optimistic and pessimistic safe set, to stay safe during exploration. It also introduces a novel way to ensure that the algorithm does not get stuck in a safe area by detecting this case and more aggressively expanding the set of considered exploration actions.

**Limitations And Societal Impact:**

The assumptions are clearly stated, and this gives some idea of the limitations of the algorithm. However, further limitations were not really discussed. Line 183 gives a worst-case scenario for sample efficiency. However, some other limitations are that this paper uses deterministic state transitions, and that it uses a discrete MDP.

Some relevant questions:
- Is there really no way for an agent to get "stuck" or accidentally visit an unsafe state as long as the assumptions hold?
- Are these assumptions valid for most robotics problems? Are there some situations that are more likely to invalidate the assumptions than others?
- How would the technique deal with limited sensor information (finite range, obstacles, a LIDAR that was not 360 degrees, etc.)?

**Main Review:**

# Originality
The approach is novel. The paper draws on unexpected sources to develop new theoretical guarantees on the algorithm's performance. ESTE is a clever workaround to getting stuck during exploration.

# Significance
A number of techniques have tackled this problem in the past with varying degrees of success (Sui et al. 2018, Wachi and Sui 2020, Berkenkamp et al. 2016 are a few notable examples). This paper is significant in the field because it seems to provide a comprehensive framework for dealing with safe RL problems - as long as the problems satisfy this (fairly relaxed) set of assumptions, the technique should apply broadly - even if the safety function has discontinuities, which is common in real world scenarios.

# Quality
This paper builds up a new safe exploration technique with logical assumptions and clearly shows the approach's effectiveness. So much of the paper hinges on these assumptions, so it is good that they are stated clearly.

It's unclear why the approach was not compared directly to the algorithms used in prior work. Many of these works even have open-source code available, so the experiments could have been true head-to-head rather than "stepwise approaches as in [36]" (line 310) which implies a re-implementation, or a similar-but-not-identical algorithm.

From my reading of the paper, it seems that the ESTE version should perform not worse than the non-ESTE version in all cases. However, in Figure 2, the ESTE version of the algorithm did seem a bit slower than the non-ESTE version, and it's unclear why this would be the case. Perhaps the difference is random deviation, however, since the experiment is fully controlled, it should be possible to do a direct, deterministic, head-to-head comparison between ESTE and non-ESTE versions of the algorithm on identical problems.

It is odd that Algorithm 1 gives the next state directly, rather than the action to be taken. This makes it difficult to see how it would extend to a situation with non-deterministic transition functions.

Is one of the baselines algorithmically equivalent to SPO-LF but with no sensor data? This would be an interesting comparison to make, to see how vital the sensor data is.

# Clarity
The paper is written and organized superbly.

Minor edits/questions:
- L160: Should $Z_\epsilon^n$ be just $Z^n$? or does $Z_\epsilon(X) = Z(X, \epsilon)$?
- L204: this follows from $g(s) = ◇(s) \geq h$? It would be nice for this to be made explicit
- L204: this only holds for $s \in S \setminus \Psi$, correct?
- L58: resprcively

**Time Spent Reviewing:**

2.5

---

> ### Author Response · Authors · 2021-08-08
> **Authors' Response**
>
> We appreciate the reviewer for helpful and thoughtful comments. We first address the reviewer’s concern in the main review and then answer the questions raised by the reviewer.
>
> **Comparison with the baselines.** We consider that the most relevant previous work is Wachi and Sui (2020). We compared our proposed method with their SNO-MDP algorithm in Section 6.1, which is based on their GitHub source code and called it GP (state) in Figure 2. On the other hand, our problem settings are slightly different from theirs; hence, we also compared our SPO-LF with adapted versions of SNO-MDP (i.e., GP (feature), Step safe GLM) for fair comparisons. The relationship between existing algorithms and our baselines may have been unclear for readers, so we will improve the terminology or presentation in the final version.
>
> Note: We apologize for a typo in Figure 2(a): GP (context) →  GP (feature)
>
> **Difference between ETSE and non-ETSE versions.** We paid attention that every method is tested on identical problems (i.e., random seeds). When the ETSE algorithm is triggered, the agent is often required to make a large move to a safe state of great uncertainty; hence, it tends to take a larger time step until convergence in the ETSE version. Therefore, we consider that our experimental results in Figure 2(a) are reasonable.
>
> **SPO-LF without sensor data.** “SPO-LF (FOV 1 x 1)” is the SPO-LF algorithm where the agent can observe only the current state. This corresponds to the one without sensor data. We will clearly mention it in the final version.
>
> Next, we will answer the questions given by the reviewer.
>
> > “Is there really no way for an agent to get "stuck" or accidentally visit an unsafe state as long as the assumptions hold?”
>
> Answer. We consider that the agent will neither get stuck nor visit unsafe states “with a high probability” as long as the assumptions hold, as guaranteed in Theorems 1 and 2.
>
> > “Are these assumptions valid for most robotics problems? Are there some situations that are more likely to invalidate the assumptions than others?”
>
> Answer. In RL problems, final layers of neural networks are often linear, which can be interpreted as a combination of feature encoding and linear mapping (Song et al., 2016). This paper incorporates “generalized” linear models (GLMs), and the assumptions made in this paper should be valid for most robotics problems. If there are cases where it is impossible or extremely hard to find an appropriate feature mapping function, the assumptions may be invalidated (This is a difficult situation for other methods as well, though).
>
> > How would the technique deal with limited sensor information (finite range, obstacles, a LIDAR that was not 360 degrees, etc.)?
>
> Answer. In our experiment, we have already dealt with limited sensor information. More specifically, the sensor range was finite and the only information in front of the agent was observed (we would like to ask the reviewer to see Figure 5 in the supplementary material if possible). In our algorithm, states that have not been observed even a single time are NOT categorized in the pessimistic safe space; hence, the agent is simply prohibited to visit a state that has never been observed.
>
> Finally, please let us answer the minor questions.
>
> **L160.** Thank you for pointing it out. We should have denoted $Z_\epsilon (X) = Z (X, \epsilon)$. In the final version, we will fix this.
>
> **L204.** Lemma 3 is for $s \in \mathcal{S} \setminus \Psi$. For the states outside $\Psi$, the agent has not observed features even a single time; hence, the confidence intervals outside $\Psi$ (see Lemmas 2 and 3) are much looser than those inside $\Psi$ (see Lemma 1). Due to the page limit, we wrote proofs (and additional explanations) in the supplemental material. We will try to include them in the main paper in the final version.
>
> We will modify our paper on the basis of the reviewer’s feedback. In Particular, we will enrich the discussion of the limitations of our algorithm. In that process, we appreciate that the questions raised by the reviewer are quite useful.
>
> References
> - Song, Zhao, et al. "Linear feature encoding for reinforcement learning." In NeurIPS. 2016.
> - Wachi, A. and Sui, Y. (2020). Safe reinforcement learning in constrained Markov decision processes. In ICML

---

> > ### Comment · Reviewer_E6K1 · 2021-08-28
> > **Acknowledgement**
> >
> > Thank you for the clarifications.

---

### Official Review · Reviewer_XEQB · 2021-07-16

**Rating:** 8
**Confidence:** 4

**Summary:**

This work proposes a novel safe RL algorithm based on locally available far-sighted features based on GLM assumption.

**Ethical Concerns:**

The reviewer believes there is no significant ethical concern in this work.

**Limitations And Societal Impact:**

The reviewer wanted to see the discussions on discretization accuracy vs the usefulness of local feature for continuous control problems.  It would be possible to discuss limitations more.

Social impact is discussed.

**Main Review:**

The idea of shared representations for safety/reward is interesting.
The paper is well written.

Major concerns:
1) in line 155, Y_return(X,bar{X}) is defined by bar{X} \cup ---.  Is it obvious that from state in bar{X} to bar{X} is feasible?  is there any action choice that makes the agent stays in the same state?  also in line 159, is Y_epsilon(X) needed in the definition of Z? (removing this term does not seem to affect the set)


Minor concerns:
1) in line 89, the definition of psi should be psi:S -> 2^S ?
2) this work considers finite sets of state and action, and also assumes that the transition is known; therefore, saying that existing GP-based method is not scalable while the proposed work is scalable is a bit misleading; what if the work is extended to unknown transition?

-----------
Review updated after the initial response

**Time Spent Reviewing:**

6 hours

---

> ### Author Response · Authors · 2021-08-08
> **Authors' Response**
>
> We thank the reviewer for valuable feedback and comments. We answer the points below:
>
> ### Major concerns.
>
> Thank you for the detailed comments. As for the $Y_\text{return} (X, \bar{X})$, we implicitly assume that the “stay” action is always feasible. We will clearly state that in the final version. Also, as for the $Y_\epsilon (X)$, we consider that it is needed in the definition of $Z$. Without $Y_\epsilon (X)$, states that do not satisfy the safety constraint may be contained in $Z$.
>
> ### Minor concerns.
> - **Definition of $\psi$.** Thank you for the suggestion! We will try to rewrite the paper based on the reviewer’s suggestion and would like to consider which definition of \psi is more reader-friendly.
>
> - **Existing GP-based method.** In the Introduction, we discussed the difference between our algorithm and the ones in the most relevant papers that deal with finite state and action spaces and deterministic transition (e.g., Turchetta et al. (2016) and Wachi and Sui (2020)). However, given there are many other GP-based safe RL approaches, our claims might be a bit misleading as the reviewer pointed out. We will carefully rewrite misleading statements in the final version.
>
> - **Unknown transition.** As denoted in the footnote on page 3, Biyik et al. (2019) well addressed the safe exploration problem with unknown transition. We consider that our method could be extended to an unknown transition model using their ideas. In our paper, to clearly convey the key ideas behind our algorithm, we determined to deal with known state transitions for simplicity.
>
> We will improve our paper in the final version based on the reviewer’s comments while discussing the limitations of our approach.
>
> References
> - Turchetta, M., Berkenkamp, F., and Krause, A. (2016). Safe exploration in finite Markov decision processes with Gaussian processes. In NeurIPS.
> - Wachi, A. and Sui, Y. (2020). Safe reinforcement learning in constrained Markov decision processes. In ICML
> - Biyik, E., Margoliash, J., Alimo, S. R., and Sadigh, D. (2019). Efficient and safe exploration in deterministic Markov decision processes with unknown transition models. In IEEE ACC.

---

> > ### Comment · Reviewer_XEQB · 2021-08-19
> > **After initial response**
> >
> > Thank you for the response;
> > Re definition of $Z$, the reviewer might be missing some, but it seems that it is the same even without having $\cap Y_{\epsilon}$? Anyway, having this term does not harm the result.  Maybe adding some figure illustrating the relation of the sets is helpful.
> > Also, given that the author(s) will modify the possibly misleading statement about the comparison to existing GP based method, the reviewer raises the score to 8 from 7.

---

> > > ### Author Response · Authors · 2021-08-19
> > > **Thank you for additional feedback**
> > >
> > > We sincerely thank the reviewer for reading through our response and raising the score. We believe that the reviewer's valuable comments are very helpful for improving our paper. We will surely modify the paper based on the reviewer's comments. Thank you again for the thoughtful comments.

---

### Official Review · Reviewer_N2r5 · 2021-07-17

**Rating:** 6
**Confidence:** 3

**Summary:**

This paper proposes SPO-LF, a safe learning algorithm for sequential decision making. The key idea is to predict a pessimistic safe space and an optimistic safe space based on the confidence intervals, and constrains the exploration in the safe space. The proposed algorithm is evaluated in the mini-grid and the safety-gym environments, compared with several baselines, and claimed to be able to handle large-scale problems, and behave more safely than the CMDP-based deep RL algorithms.

**Limitations And Societal Impact:**

I did not find where the limitations are discussed. Are there any limitations of the proposed method?
I did not find any potential negative societal impact of this work.

**Main Review:**

Safe exploration is critical to apply reinforcement learning in real-world systems, such as robotics and self-driving cars. This paper tries to address this important challenge by extending the dynamic programming approach to ensure zero unsafe actions during training while still learning near-optimal policies.

My major concern of this paper is its unsupported claims that "our algorithm is more applicable to large-scale problems" and "behaves more safely than existing advanced deep RL methods with constraints". It appears to me that SPO-LF only works for environments with discrete state and action spaces. Discretizing the state and the action space, as did in the safety-gym evaluation, is not scalable to large scale continuous control problems. For this reason, comparisons with CPO, PPO-Lagrangian or TRPO-Lagrangian are not fair because the baseline algorithms are solving a much more challenging continuous-space problem.

Here are a few clarification questions:
1) Equations below line 121. Why is it necessary to have \phi_{s_\tau} right before "=" on the left hand side of equations?
2) It would be great to give some intuitions of Assumption 2 "The link function satisfies ..." What's its physical meaning? How strong is this assumption?
3) I had some trouble understanding line 223-225. What is the physical meaning of C_t?
4) The ETSE seems to be a heuristic for exploration whose goal is to reduce uncertainty. I could misunderstand here, but ETSE can potentially make the entire algorithm unsafe because it needs to explore the states with large uncertainty. If this is the case, I am surprised to see that the number of unsafe actions is still zero for both the grid world and safety-gym evaluations. Did I miss anything here?

**Time Spent Reviewing:**

3 hours

---

> ### Author Response · Authors · 2021-08-08
> **Authors' Response**
>
> We thank the reviewer for the considerable comments. First of all, we would like to address the concerns raised by the reviewer.
>
> **Major concerns.** Please let us emphasize that our claims are that our proposed SPO-LF is
> more efficient in terms of sample complexity and computational cost and more applicable to large-scale problems than previous safe RL methods with theoretical guarantees
> comparably sample-efficient and safer compared with existing advanced deep RL methods with safety constraints.
> Please be noted that we do NOT claim that our proposed method is more scalable than CMDP-based deep RL methods (e.g., CPO, PPO-Lagrangian). We claim that our proposed method is more scalable than previous safe RL methods “with theoretical guarantees”, which is supported by our experiment in Section 6.1. Safe RL methods with theoretical guarantees have suffered from a lack of scalability. Specifically, even a 150 x 150 grid world cannot be solved in a reasonable time as shown in Figure 3. Indeed our proposed method may be less scalable compared with advanced deep RL methods, but it is a solid technical step that our algorithm can be applied to a much larger problem while maintaining the theoretical guarantees.
>
> We believe that our proposed method compensates for the shortcomings of safe RL methods with theoretical guarantees (e.g., Turchetta et al. (2016), Wachi and Sui (2020)) and advanced deep RL methods (e.g., CPO), which potentially sets out a research direction to bridge the gap between two distinct methods. Our experiments show that our proposed method makes up for the shortcomings of both, and we consider that our claims have been fully supported by our two experiments.
>
> Next, we will answer the questions.
>
> **Equation for maximum likelihood estimator.** Since observations for reward and safety functions contain noises, we consider predicting unknown coefficients $\theta_r$ and $\theta_g$ as maximum likelihood estimators (MLEs). In our paper, based on Filippi et al. (2010) and Li et al. (2017), we first consider the log-likelihood function written as
>
> $\Sigma_{\tau=0}^t (r(s_\tau) \phi_{s_\tau}^\top \theta - b(\phi_{s_\tau}^\top \theta) + c(r_\tau) )$,
>
> and then get the MLE by differentiation. In the above equation, $b = \dot{\mu}$ and $c$ is a real function. In this calculation process, $\phi_{s_\tau}$ right before "=" on the left-hand side of equations appears. This part might be hard to understand for readers. In the final version, we will add more explanations and pointers.
>
> **Assumption 2.** This assumption is on the local behavior of the first-order derivatives of the link function near $\theta^*$. Specifically, we assume that the first-order derivative of $\mu$ is more than zero around $\theta^*$. This assumption is needed for the sake of stable calculation of MLEs (if $\dot{\mu}(\phi^\top \theta)$ = 0, asymptotic variance of $\phi^\top \theta$ will go to infinity). Also, we think that this is a mild assumption. Specifically, when the link function is linear or Poisson, this assumption is automatically satisfied.
>
> **Meaning of $C_t$.** $C_t$ is the confidence interval of the predicted safety function, which is guaranteed not to expand over iterations. We used $C_t$ rather than $Q_t$ for a technical reason; that is, we need to ensure that the pessimistic safe space does not shrink over iterations. This technique has been commonly used in previous work (e.g., Sui et al (2015), Turchetta et al. (2016)) as well.
>
> **How to deal with safety in the ETSE algorithm.** As written in Equation (5), the next state is always chosen within $\mathcal{X}_t^-$; hence, the agent will visit only a pessimistically identified safe state. In other words, Theorem 1 is still valid even during the ETSE phase. That is the reason why the number of unsafe actions is zero in our experiments.
>
>
> References
> - Li, L., Lu, Y., and Zhou, D. (2017). Provably optimal algorithms for generalized linear contextual bandits. In ICML.
> - Filippi, S., Cappe, O., Garivier, A., and Szepesvári, C. (2010). Parametric bandits: The generalized linear case. In NeurIPS.
> - Sui, Y., Gotovos, A., Burdick, J. W., and Krause, A. (2015). Safe exploration for optimization  with Gaussian processes. In ICML.
> - Turchetta, M., Berkenkamp, F., and Krause, A. (2016). Safe exploration in finite Markov decision processes with Gaussian processes. In NeurIPS.
> - Wachi, A. and Sui, Y. (2020). Safe reinforcement learning in constrained Markov decision processes. In ICML.

---

> > ### Comment · Reviewer_N2r5 · 2021-08-30
> > **Thanks for the response**
> >
> > Your response partially addressed my major concern. Your answers to my questions make sense.
> >
> > I agree with you that Section 6.1 fully supports the claim that SPO-LF is more scalable than previous safe RL methods with theoretical guarantees. I also acknowledge that "it is a solid technical step that SPO-LF can be applied to a much larger problem while maintaining the theoretical guarantees."
> >
> > However, I still think that "it is safer compared with existing advanced deep RL methods with safety constraints" is an over-statement. It appears to me that these CMDP methods (CPO, PPO-Lagrangian and TRPO-Lagrangian) can solve large-scale problems that SPO-LF cannot. So SPO-LF is only safer in a subset of problems that CMDP methods can solve. I believe that a more fair claim should be "the proposed method is safer compared with existing advanced deep RL methods with safety constraints for a subset of problems with discrete state and action spaces."
> >
> > While I agree that this paper made a step pushing the frontier, as a robotic researcher, I still want point out that the paper makes strong assumptions and may not be scalable to handle real robots (e.g. locomotion).
> >
> > For above reasons, I bumped up the score from 4 to 6.

---

> > > ### Author Response · Authors · 2021-08-30
> > > **Thank you for the additional comments**
> > >
> > > We would like to express our sincere gratitude to the reviewer who read through our responses and raises the score. We believe that the valuable comments from the reviewers will be very helpful for us to improve our paper. We plan to revise our paper based on the reviewers' comments so that each statement is fair. Thank you very much for your considerable comments.

---

### Decision · Program_Chairs · 2021-09-27

**Decision:**

Accept (Poster)

**Comment:**

All reviewers recommend acceptance, two strongly. The reviewers recognize the importance of the topic (safe exploration in reinforcement learning) and the significance of the submission within this specific area. Any concerns were addressed clearly by the authors, and so the AC recommends acceptance.

Note: There appears to be some confusion surrounding a claim "our algorithm is more applicable to large-scale problems than previous safe RL methods with theoretical guarantees". The point of confusion appears to be what constitutes a previous safe RL method with theoretical guarantees. If CPO is included (it is safety-inspired and has some theoretical guarantees [though not strong guarantees of safety]) then this may not be true. However, when comparing to algorithms with very strong (high-confidence) safety guarantees, the author's claim is clearly true. The AC hopes that the authors do not dismiss the reviewers concerns about this line, but rather aim to clarify this line to ensure that other readers do not have the same misconception about the authors' intent with the statement.